# Emergence of the Fungal Rosette Agent in the World: Current Risk to Fish Biodiversity and Aquaculture

**DOI:** 10.3390/jof9040426

**Published:** 2023-03-29

**Authors:** Rodolphe Elie Gozlan, Marine Combe

**Affiliations:** ISEM, Université de Montpellier, CNRS, IRD, 34090 Montpellier, France

**Keywords:** food security, aquatic conservation, disease, *Sphareothecum destruens*, fungi, invasion, outbreaks, healthy carrier

## Abstract

The emergence of pathogenic fungi is a major and rapidly growing problem (7% increase) that affects human and animal health, ecosystems, food security, and the economy worldwide. The Dermocystida group in particular has emerged relatively recently and includes species that affect both humans and animals. Within this group, one species in particular, *Sphareothecum destruens*, also known as the rosette agent, represents a major risk to global aquatic biodiversity and aquaculture, and has caused severe declines in wild fish populations in Europe and large losses in salmon farms in the USA. It is a species that has been associated with a healthy carrier for millions of years, but in recent decades, the host has managed to invade parts of Southeast Asia, Central Asia, Europe, and North Africa. In order to better understand the emergence of this new disease, for the first time, we have synthesized current knowledge on the distribution, detection, and prevalence of *S. destruens*, as well as the associated mortality curves, and the potential economic impact in countries where the healthy carrier has been introduced. Finally, we propose solutions and perspectives to manage and mitigate the emergence of this fungus in countries where it has been introduced.

## 1. Introduction

Emerging fungal pathogens pose a growing threat to global health, ecosystems, food security, and the world economy [1]. Between 1995 and 2010, the proportion of fungal infections in plants and animals recorded in the ProMED database (the Program for Monitoring Emerging Diseases) increased from 1% to 7%, and a positive trend in the proportion of fungi infecting animals and plants was observed over the period of 2007–2011 with global fungal disease alerts [1]. Furthermore, data from a meta-analysis and literature searches have shown that fungal infections remain the major cause (65%) of biodiversity loss due to pathogen [1].

Although fungi have been known for a long time to pose a widespread threat to plants, the impact of fungal infections on animal health has been underestimated until recently, with major declines seen in wildlife due to fungal emergences [1]. Indeed, in 1997, a major fungal infection of amphibians caused the largest biodiversity loss event in the world. The chytrid fungus *Batrachochytrium dendrobatidis*, co-introduced with American bullfrogs (*Lithobates catesbeianus*), was responsible for the extinction of at least 500 amphibian species in 54 countries [2,3]. In the Americas, *B. dendrobatidis* has caused a loss of more than 40% of amphibian species [4], also leading to major ecological changes [5]. Globally, chytridiomycosis (the disease caused by *B. dendrobatidis*) has led to the decline of nearly half of all amphibian species [6]. Importantly, the fungal diseases causing the decline in global populations are emerging in a wide range of terrestrial, but also wild and farmed aquatic animal species, including, for example, soft corals (see-fan aspergillosis caused by *Aspergillus sydowii*) and tilapia fish (epizootic ulcerative syndrome caused by *Aphanomyces invadans*) [1]. Similarly, the emerging fungus *Aphanomyces astaci* has caused a dramatic decline in freshwater brown crayfish populations worldwide, through a disease called crayfish plague [7]. Despite their impact on biodiversity, fungal diseases are also a major threat to the aquaculture sector worldwide [8], i.e., they have rapidly expanded the range of species farmed, including crustaceans, mollusks, and finfish, notably tilapia [9], and thus, food security. For example, fungal infections are among the most common diseases in fish in both temperate and tropical areas [10]. The main fungal diseases reported in aquaculture include those caused by fungi of the genera *Aphanomyces*, *Branchiomyces*, *Lagenidium*, *Saprolegnia sp*., *Sirolipidium*, *Phoma*, *Aphanomyces invaderis*, *Leptolegnia*, and *Dictyuchus*, and infect a wide variety of farmed fish such as rainbow trout, yellowtail, mackerel, herring, flounder, cod, salmonids, tilapia, carps, etc. [10]. These observations therefore highlight the need for improved surveillance systems, detection of emerging fungal pathogens, monitoring disease prevalence, and a sound knowledge of host–pathogen distribution (geographic range) and interaction (pathogen virulence vs host susceptibility).

In 2005, a global risk of disease emergence for freshwater fish biodiversity was identified in Europe and directly linked to the accidental introduction in the early 1960s of the highly invasive Asian gudgeon, also named topmouth gudgeon *Pseudorasbora parva*. A healthy carrier of a newly described fungus, the rosette agent *Sphaerothecum destruens*, in reference to the sphere containing spore-like structures (*Sphaerothecum*), *destruens* meaning destructive, once established in the host fish, the infection causes widespread destruction of various tissues [11]. Since then, severe declines in fish populations have been confirmed across Europe, including wild and farmed fish, following the arrival of this invasive host–parasite complex [12]. *S. destruens* is a unicellular eukaryotic fish parasite and an obligate intracellular parasite known as the “rosette agent”. This parasite was originally assigned to the category “Dermocystidium-like” because it shared similar morphological features with other enigmatic parasites of fish and crustaceans [11]. Kerk et al. [13] were the first to obtain the complete DNA sequence encoding the small rRNA subunit (18S rRNA gene) of *S. destruens* and showed that it shared its most recent common ancestor with the choanoflagellates, a group of sister protists of multicellular animals. Further phylogenetic analyses of the 18S rRNA gene of two *Dermocystidium* spp., *S. destruens*, *Ichthyophonus hoferi*, and *Psorospermium haeckeli* confirmed their relationship and divergence from the animal–fungal dichotomy [14]. Based on these results, Ragan et al. [14] assigned them to the clade of DRIPs (DRIP for the first letter of each of their names), also belonging to the class Ichthyosporea [15]. Subsequently, Mendoza et al. modified this classification and created the class Mesomycetozoea [16,17], previously proposed by Herr et al. [18] and notably composed of the Order Dermocystida and the Order Ichthyophonida [15]. The class Mesomycetozoea includes 10 different parasitic and saprophytic micro-organisms belonging to the genera *Amoebidium*, *Anurofeca*, *Dermocystidium*, *Ichthyophonus*, *Pseudoperkinsus*, *Psorospermium*, *Rhinosporidium*, *Sphaeroforma*, and *Ichthyophonida* sp. [17] and the Order *Dermocystida* includes *Dermocystidium* spp., *Rhinosporidium seeberi*, and *S. destruens*, which share the ability to cause infections in animals. For example, more than 20 species of *Dermocystidium* spp. cause infections in carp, goldfish, salmonids, eels, newts, and frogs [14]. However, the more recent phylogenetic classification grouped *S. destruens* with animals, fungi, and choanoflagellates in the super-group Opisthokonta [19].

Published data indicate ancient host–parasite coevolution with multiple introduction events occurring across Europe from admixed host population sources of *P. parva* [20,21]. Furthermore, *S. destruens* is considered to be a generalist parasite with a wide host range [12,22] and high tolerance to temperature variations (from the mountainous Tcheremoch River in Ukraine to the desert wadi of Felrhir in Algeria), and several reports indicate the global spread of its healthy carrier host *P. parva*. Our objective was to synthesize the latest knowledge on the distribution, detection, and prevalence of *S. destruens* in invasive and native fish host populations, as well as to review the biology and pathology of *S. destruens* and the economic impact of this disease in wild and farmed aquatic animals.

### 1.1. Worldwide Distribution of Sphaerothecum destruens

The disease caused by *S. destruens* first emerged in salmonids in the USA and was first described in 1984 [23]. Between 1983 and 1984, 80% mortality occurred in a group of 2.5-years-old chinook salmon *Oncorhyncus tshawytscha* brood stock, reared in seawater net-pens in Puget Sound, Washington, DC, USA. These mortalities could not be attributed to previously known pathogens. The diseased fish were anemic with marked lymphocytosis and had enlarged kidneys and spleens. Light and electron microscopic examinations of the spleen and kidney tissues revealed the presence of numerous intracellular spherical organisms of 3–7 µm in size with chemical and structural characteristics similar to marine algae or fungi. The infectious agent responsible has been described as a systemic protist and has been termed a “rosette” or “chinook rosette agent” because of the clustered organization of organisms observed in the stained tissues of infected fish [23]. This first RA-1 isolate of *S. destruens* was amplified on the CHSE-214 cell line developed from Chinook salmon embryos and held in the American Type Culture Collection (ATCC, Accession Number #50643) [24]. A few years later, chronic mortalities in subadult Atlantic salmon *Salmo salar* received as eyed eggs from Finland and reared in spring water on a private farm in northern California, USA, were reported [25]. Although microscopic examinations suggested that the disease was caused by an organism similar to that described by Harrell et al. [23], it was termed “Dermocystidium-like” and a second isolate of *S. destruens* RA-2 was deposited at the ATCC (Accession Number #50644) [25]. Finally, a third emergence of the disease was reported in the USA in captive subadult and winter-run adults of *O. tshawytscha* broodstock from Sacramento River maintained at the Bodega Marine Laboratory (BML, Bodega Bay, CA, USA) in collaboration with the Coleman National Fish Hatchery (CNFH, CA, USA) and several state and federal agencies [26]. Among the 1991–1994 brood years, disease prevalence peaked at 40.1% in the 1991 year class. *S. destruens* was first detected in a few 1991 brood year salmon (14–16 months old) that had never been transferred to seawater since their arrival from the original freshwater hatchery (CNFH). Subsequently, the parasite was detected in wild adult late-fall-run *O. tshawytscha* returning to freshwater to spawn in the upper Sacramento River. Since 1994, surveillance by the California–Nevada Fish Health Center has demonstrated the presence of *S. destruens* in up to 32% of late-fall-run adult *O. tshawytscha* returning to Battle Creek on the Upper Sacramento River, suggesting the persistence of the disease in the BML and a potential risk of disease emergence. A third isolate, named RA-3 or BML strain, was isolated from *O. tshawytscha*-infected kidneys, amplified in fish cell lines, and also deposited at ATCC (Accession number #50615).

For several years, the disease caused by *S. destruens* was only reported in salmonids reared on the northwest coast of the USA (Washington State and California). However, in 2005, Gozlan et al. [27] were the first to describe the emergence of *S. destruens* disease in invasive populations of *Pseudorasbora parva* in Europe, but also in the non-native cyprinid *Leucaspius delineatus* (sunbleak) in England [27,28]. As the local declines of *L. delineatus* in Europe coincided with the initial introduction of *P. parva* into Romanian ponds in 1960 near the Danube, followed by its rapid spread throughout Europe, they hypothesized that Asian populations of *P. parva* might represent the asymptomatic carrier of *S. destruens*. Their results not only provided the first occurrence of the disease outside North America, but also showed for the first time that several species of cyprinids were also susceptible to *S. destruens* and could develop the disease, thus extending the potential host range of this parasite [27,28]. Furthermore, their experimental results confirmed that *P. parva* populations could represent asymptomatic carriers of *S. destruens*. Importantly, their work pioneered the use of targeted molecular tools to detect *S. destruens* in infected fish [22]. The RA-4 isolate of *S. destruens*, also named UK-Cefas1, was extracted from tissues (kidney and liver) of wild-infected *L. delineatus* following cohabitation experiments with *P. parva* [27]. Although no obvious morphological or pathological differences between *S. destruens* infections in *L. delineatus* and salmonids were noted, RA-4 isolate showed genetic variability compared with the three American isolates (RA-1, RA-2, RA-3) [12,28].

Following this work, based on the rapid spread of invasive *P. parva* populations in freshwater ecosystems (see Figure 1A–C), several research groups have been investigating the potential presence of the disease in European fish. In the Netherlands, populations of *P. parva* were introduced in 1992 and rapidly colonized lakes via the Meuse river [29].

Invasive *P. parva* populations sampled in 2008 in the Everlose Beek floodplain and sampled in 2012 in the Teelebeek stream (floodplain of the Meuse river) were also found to be infected with *S. destruens* [30,31]. In Turkey, from 2009 to 2013 in the Sariçay stream of Mugla, three species of native fish *Oxynoemacheilus* sp. (Nemacheilidae), *Petroleuciscus smyrnaeus*, *Squalius fellowesii*, and the non-native *Lepomis gibbosus* (Percidae) as well as farmed seabass *Dicentrarchus labrax* (Moronidae) cohabiting with invasive wild populations of *P. parva* were found to be infected by *S. destruens*. Kidney, liver, spleen, and gonad tissues were all tested for the presence of the parasite using PCR assays and histological examinations, confirming its wide host range as well as its broad cellular tropism [32]. In France, invasive populations of *P. parva* were first reported in 1980 [33] and a survey of fish populations conducted between 1990–2009 showed that this non-native species has spread dramatically [34]. In 2016, Charrier et al. [35] showed the presence of *S. destruens* in 12 individuals of *P. parva* caught in a small tributary stream of the Adour River near Dax, France [35]. Just one year later, Boitard et al. [36] reported the occurrence of two additional natural infections of salmonids in France. In November 2015, chronic mortality occurred in brown trout (*S. trutta*) and rainbow trout (*O. mykiss*) at an experimental facility, followed in 2016 by a second unusual episode of fish mortalities at a rainbow trout farm, both located in southwest France. In both outbreaks, kidneys, livers, and spleens were collected for histological and molecular examination and revealed the presence of foreign cells 2–4 µm in diameter with eosinophilic inclusions and were observed to be single or arranged in rosettes (spores aggregates) consistent with *S. destruens*. Sequencing of PCR products revealed a disease prevalence of 80% and 100% in *O. mykiss* and *S. trutta*, respectively [36]. In France, this pioneering work led to the deployment of a nationwide project involving seven French departmental angling associations and two research institutes. Sampling took place across the country between 2017–2019 and included 10 freshwater sites from which 50 invasive individuals of *P. parva* and other native species were collected and screened by PCR for the presence and prevalence of *S. destruens* [22]. The results showed a wide distribution of *S. destruens* in freshwater sites in France with at least five out of 10 of the sampling sites, such as Ain (central-eastern France), Indre (central France), Gironde (south-western France), Bouches-du-Rhône, and Corsica island (south-eastern France), where *S. destruens* DNA was detected in five native fish species and one *P. parva* population [22]. Furthermore, it confirmed the wide range of climates (from temperate to oceanic and Mediterranean) and habitats (lotic and lentic) that *S. destruens* can tolerate. Other invasive populations of *P. parva* have been found to be infected and carrying *S. destruens* in the UK and Spain with a prevalence of 5% in both populations [37].

Finally, the third region of the world where *S. destruens* has been reported to date is among native *P. parva* populations in China [37], and more recently, in an invasive population in Lào Cai Province, North Vietnam (Gozlan pers. Com.). Indeed, Sana et al. [37] used Gozlan’s extensive 2010 sampling campaign of *P. parva* populations in its native and invasive range to detect the presence of the parasite by PCR assays targeting the 18S rRNA gene. In the native range, among 10 Chinese populations collected in different locations [20], they were able to detect the presence of *S. destruens* in nine populations with a prevalence ranging from 0 to 10% depending on the population tested and an overall prevalence of 6%. By analyzing the ITS-1 region of geographically distinct *S. destruens* isolates, [37] showed a clustering of isolates according to their geographic origin. British and Chinese *S. destruens* isolates clustered together, Turkish isolates were distinct, although more closely related to British and Chinese isolates, and American isolates formed another clade.

Taken together, these results show that the presence of *S. destruens* is mainly linked to the presence and introduction of the healthy carrier *P. parva*. It is therefore logical to assume that where *P. parva* was introduced, *S. destruens* has also been introduced, although not all local populations have been formally tested yet. Previous studies have suggested coevolution between *P. parva* and *S. destruens* lineages [32,37], indicating a true coexistence of several million years [20]. Finally, Combe and Gozlan [12] suggest that the origin of the US strains is to be sought among infected Asian O. tshawytscha (eastern Russia, Japan) living in sympatry with native *P. parva* populations, which would have contaminated their American counterparts during their migratory movement in the North Pacific. Indeed, Arkush et al. [26] reported that 33% of returning American Chinook stock from a Californian river were positive for *S. destruens*. Therefore, it is expected that the American strains will be closely related to Japanese strains when the latter are tested.

### 1.2. Mortalities Associated with the Emergence of S. destruens

The parasite *Sphaerothecum destruens* causes low-level mortalities, which makes it difficult to identify in wild fish populations. In many cases, whether in a pond, lake, or watershed setting, few will be aware of the mortalities that follow the arrival of the healthy *P. parva* carrier [12,38]. First of all, the water is not transparent, which makes it difficult to see a dying fish that is not on the surface, and secondly, few people observe aquatic ecosystems on a daily basis [38]. It is not as in the case of a virus or pollution where we would see mass mortalities on the surface that would sound the alarm for a reaction from competent authorities. In the case of the emergence of the agent, the mortalities are more insidious, with low-level mortalities adding up to a decline in the target populations [27,38,39]. The observer interested in a target population will see fish that look healthy, but there will be fewer and fewer of them until there is a decline that may be total [39]. This has been the case for sunbleak *Leucaspius delineatus* populations, which have disappeared from the whole of Europe with a few residual pockets that would merit a targeted and rapid conservation plan [27,40]. Locally, populations of cyprinids and salmonids have declined significantly or even disappeared (refs). In this context, it is not surprising to see that the first identification and isolation of *S. destruens* was observed in aquaculture systems [24,26], where it is indeed easier to monitor mortalities and therefore, to perform autopsies in order to identify the agent causing these mortalities [11]. When *P. parva* is introduced into an aquatic ecosystem where susceptible species are present, there will be an initial release of spores, probably in the feces or during reproduction. These spores in contact with fresh water will produce zoospores which will then contaminate other fish. Contamination by predation may also occur. In the first few weeks after introduction into a pond, no mortality will be observed and then, with time, the susceptible species will be less and less abundant without any mortality being observed from the edge of the pond. This can lead to complete extinction. In small aquaculture ponds, it is easier to observe mortalities, as was the case in the USA with salmonids.

The emergence of *S. destruens* in the different species tested, whether in aquariums, ponds, or at the level of a catchment area, shows a very similar mortality curve for susceptible species with the majority of mortalities occurring in the first two months and then spreading out over time towards a spiral of extinction [27,32,39]. However, from what studies have been able to show ([27,39,41], Figure 2), in an artificial environment, mortality in the first month averages around 47% and reaches 54% in the second month, whereas in a natural environment or one subject to natural conditions, such as a pond, mortality averages 89% in the first month and climbs to 92% after two months (Figure 2 [32]). Therefore, there is probably an environmental effect on the kinetics of infection-related mortalities due to *S. destruens*. Andreou et al. had shown the impact of temperature on the production and longevity of rosette agent zoospores, with zoospore production being more spread out at low temperatures (i.e., 4 °C) and faster at high temperatures (i.e., 30 °C). However, the number of zoospores produced remains the same overall, regardless of temperature. Additionally, in the natural environment, the physiological state of the hosts and their level of stress, which has an indirect impact on the efficiency of the immune system, can also explain these differences in the mortality curves (Figure 2). Indeed, aquarium fish are fed ad libitum with little competition for resources and no risk of predation. However, we must bear in mind that the impact of the agent on a population of a sensitive species may eventually lead to a total decline in the population, as has been the case, for example, in salmon aquaculture in the USA, where *O. tshawytscha* and *S. salar* stock has been decimated, on the *L. delineatus* in semi-natural basins [27], in reservoirs with the disappearance of rudds following the introduction of the agent *S. destruens* [22], or even on a catchment scale [32]. It is therefore extremely important, after the introduction of the healthy carrier *P. parva* into an aquatic system, to test to confirm the presence of *S. destruens* and to quickly take the necessary isolation measures in order to limit the extent of mortality that is induced to other populations. At the same time, some mortalities, such as those in USA aquaculture that were not related to the introduction of the healthy carrier reached mortality rates close to those related to the introduction of *P. parva*, with the major difference being that the agent rapidly disappeared from these systems, which is the opposite of what happened when the healthy carrier was present since healthy carriers also allow for the persistence of the parasite in the system (role of reservoir). This indicates the crucial indirect role played by *P. parva* on *S. destruens*-related mortalities.

### 1.3. Biological Characteristics of S. destruens and Associated Disease Pathology

The life cycle of *S. destruens* and its mechanisms of entry into cells, replication, and transmission still remain poorly documented. *S. destruens* is an obligate intracellular (intracytoplasmic) eukaryotic parasite. Its life cycle consists of (i) a spherical spore stage observed both in in vitro cell cultures and infected fish tissues, ranging in size from 2–4 µm in diameter for undivided stages and 4–6 µm in diameter for dividing stages [11,26] and (ii) a motile, uniflagellate zoospore stage experimentally reached after three days of incubation-free spores at 15 °C in freshwater (distilled water) and comprising a body of about 2 µm and a flagellum of about 10 µm in length [11]. Spores divide by partitioning and are therefore thought to replicate asexually by fission of the mother cytoplasm and its organelles to generate at least five daughter cells [11,26]. Once released, each spore may infect other adjacent tissues or be excreted primarily through bile, urine, gut epithelium and, to a lesser extent, through gills and seminal and ovarian fluids [11,26]. Fish infection is thought to occur by ingestion and gut penetration, or by attachment to the gills or skin [11] (Figure 3). The flagellum of the zoospore forms a coil around the body before uncoiling, resulting in a motile zoospore propelled forward by undulatory movements, potentially allowing it to actively reach a new host [11], as has been demonstrated for the *Rhinosporidecae* member *Dermocystidium salmonis* [42], but also for pathogens with free-living infectious stages generally capable of infecting multiple hosts [43]. Harrell et al. [23] showed that the growth of the pathogen was inhibited at 5–10 °C while the number of infected host cells and the number of produced *S. destruens* spores significantly increased at 20 °C. Later, Andreou et al. [44] experimentally showed that zoospores could have periods of inactivity between bursts of activity and found that a water temperature of 15 °C was optimal for the production (in terms of number) of zoospores and the duration of zoosporulation (up to 18 days). In contrast, increasing water temperature (above 15 °C) decreased the concentration of zoospores and the duration of zoosporulation, suggesting that cell-free spores can remain viable in freshwater for long periods before zoosporulation [44].

### 1.4. S. destruens Ultrastructure

In all histological analyses, *S. destruens* spores appeared pink to red when stained with eosin (staining for basic and acidophilic proteins within and between cells), PAS-positive (Periodic Acid Schiff), argyrophilic (Warthin–Starry and Grocott’s), and basophilic (Giemsa staining), but not acid-fast (Ziehl–Neelsen) [26]. *S. destruens* spores are composed of a well delineated trilaminar cell wall which is coated by a dense fibrogranular layer that forms the partitions between dividing daughter cells [25]. The *S. destruens* cell wall is separated from the host cell cytoplasm by an intermediate amorphous region and an electron-dense layer with another membrane originating from the host cell [45], potentially suggesting that *S. destruens* enters the host cell by endocytosis or phagocytosis. Spores contain ribosome-laden cytoplasm with scattered segments of rough endoplasmic reticulum, numerous vesicular mitochondria, and a single nucleus (Figure 3). Numerous vacuoles sometimes containing concentric bodies, electron-dense bodies ranging in shape from spheres to rods-like, and Gram-positive lipid droplets were also present in the cytoplasm [25,26,45]. These intracytoplasmic materials and spore structures were observed for both stages, i.e., for dividing and non-dividing RA spores [26]. Furthermore, these ultrastructural features of *S. destruens* spores appear to be similar in salmonids and cyprinids, such as *S. salar* [25], winter-run *O. tshawytscha* [23,24,26], and *L. delineatus* [45].

Histopathology is associated with *S. destruens* infection. Although Harrell et al. [23] were the first to describe *S. destruens* disease in *O. tshawytscha*, Elston et al. [24] were the first to isolate *S. destruens* (using CHSE-214 cell line) from infected fish and validated the realization of Koch’s postulates. At that time, they were able to describe the intracellular localization of *S. destruens* in macrophages and endothelial cells and confirmed the intracellular nature of the parasite by electron microscopy. They also observed focal areas of *S. destruens* proliferation in the spleen and kidney with necrosis of adjacent tissues [24], suggesting that intracellular replication of *S. destruens* ultimately leads to host cell death [26]. Internal examination of *S. salar* revealed widely disseminated nodules in the kidney, spleen, liver, and gonads, with splenic and hepatic lesions characterized by granulomas surrounded by multiple layers of fibroblastic cells and macrophages containing numerous parasites [25]. Although *S. destruens* spores were mainly observed within macrophages (and to a lesser extent within host cells), cell-free *S. destruens* spores have also been reported [25]. Signs of disease in moribund fish were unremarkable, with the exception of some fish with advanced infection that were slightly emaciated, but did not appear to be anemic, as evidenced by the absence of gill and blood pallor despite the involvement of hematopoietic tissues [25].

These preliminary histopathological examinations of infected tissues from naturally infected North American salmonids were later confirmed in captive broodstock of Sacramento River winter-run *O. tshawytscha* [26] and experimentally infected *S. salar* [46]. The authors reported two different forms of microscopy lesions in infected fish: (i) a nodular (focal) form of the disease characterized by distinct multifocal granulomas that replaced the normal parenchyma in the kidney, liver, and spleen. The granulomas were well delineated from the normal parenchyma and were characterized by central cores of necrotic material or closely apposed macrophages. Thin rims of fibroblasts sometimes surrounded the granulomas, while aggregates of *S. destruens* spores were often seen in the central areas and in or between the macrophages of the granulomas. This nodular form, characteristic of fish with an immune response and therefore able to contain infections, was mainly observed in visceral organs such as the liver, kidney, and spleen, but also in the heart, the mesentery surrounding the intestinal tract, and between the pyloric cecae [26]; (ii) a disseminated form of the disease with widely dispersed *S. destruens* spores in various organs and cells, including kidney, liver, spleen, heart, gills, brain, ovary, testes, and hindgut, as well as hematopoietic, epithelial, and mesenchymal cells [26,46]. This form was characteristic of fish more susceptible to *S. destruens* with the absence or low expression of host cell immune responses [25,26]. *S. destruens* spores were observed both in intra- and extracellular locations in tissues and as single rosettes or as aggregates of 4–5 rosettes. Disseminated infections included enlargement and pallor of the liver, kidney, and spleen and showed little macrophages or fibroblast proliferation around the lesions, while areas of oedema and focal necrosis were present in close proximity to *S. destruens* spore proliferation [26]. In the disseminated form of *S. destruens* disease, kidney tissues were characterized by necrosis, loss of tubules, membranous glomerulonephritis, and necrotizing intestinal nephritis. Single or rosette structures of *S. destruens* spores were observed in the cytoplasm of the biliary epithelium and renal tubules and in the lumen of bile ducts, suggesting that bile and urine might be routes of *S. destruens* excretion. Necrosis of the renal tubular epithelium and multifocal hepatocellular necrosis were also reported. Spleen tissues contained numerous *S. destruens* spores, single or in aggregates, in the pulp spaces of the spleen and in the cytoplasm of sinusoidal macrophages and reticuloendothelial cells. Rosettes of *S. destruens* were found in the lumina and tunicae media of splenic arterioles where they were accompanied by segmental necrotizing vasculitis. In early infections (subadult fish), *S. destruens* was sometimes observed in the gill vessels, while in advanced infections (adult fish), it was often found in subserosal aggregates in the swimbladder, mainly in macrophages, regardless of the type of lesions. It has sometimes been reported that *S. destruens* was found in the epidermis, urine, seminal and ovarian fluids, and in the mucosa of the intestine, suggesting that the gut epithelium, skin, and gills could represent a second route of *S. destruens* excretion ([26] Table 1).

The disease and pathology observed in wild populations of infected *L. delineatus* (*Cyprinidae*) in the UK (Stoneham Lakes) were similar to those reported in *O. tshawytscha* from the USA, although some authors found slight differences, such as the presence of *S. destruens* spores in giant cells and the observation of only the smallest (2–4 µm) spore morphology [28,45,46]. The disseminated form of the disease was most frequently observed in infected *L. delineatus* (80% of infected fish) [28,45]. Numerous stages of RA spores were observed with sizes ranging from 2–4 µm in diameter and most stages were intracellular. As in salmonids, *S. destruens* infection induced hepatocellular necrosis [28] and an inflammatory response in the testes and the liver, involving an influx of phagocytic cells with some lymphocytic infiltration of the liver parenchyma and necrosis [45]. In ocular tissues, *S. destruens* spores have been found in macrophages and giant cell formation has been reported. Various stages of granulomas have been described, ranging from enlarged macrophage aggregates surrounded by a single-cell layer of connective tissues to well demarcated lesions surrounded by a thick fibroblast layer [45].

### 1.5. Comparisons with Other Closely Related or Fungal Parasites

Although members of the order Dermocystida share the ability to cause infections in animals, their phenotypic characteristics and the diseases they cause are very different [11]. Their life cycles are not well known and evidence of sexual development has not yet been described [11]. Members of the genus *Dermocystidium* develop numerous spores 5–8 µm in diameter contained in cysts (a sac-like pocket) that average 0.5–1.1 mm in size and are localized between the epithelial (extracellular) tissues of the fish host without the expression of a host inflammatory response [42]. In *Dermocystidium salmonis*, once the spores mature, they differentiate into multiple flagellated zoospores that are about 1 µm in diameter, which are released from the cyst and are then able to reinfect the gill epithelium of a new fish host [42]. *Dermocystidium cyprini* is also known to have a flagellated zoospore stage in its life cycle [47]. The existence of a zoospore stage in other Dermocystidium species has not been reported. Interestingly, the ultrastructure of *S. destruens* has been found to be quite similar to *Dermocystidium macrophagi* infections in rainbow trout *Oncorhynchus mykiss* [48] and *Dermocystidium sp*. described from *S. trutta* and *S. salar* cultured in Ireland [49], although the “signet ring” appearance of a prominent vacuole previously described for *Dermocystidium* sp. [50] has not been observed in *S. destruens* spores infecting winter-run *O. tshawytscha* [11] or *S. salar* [25]. Members of the genus *Rhinosporidium* have a different life cycle to the genus *Dermocystidium* and *S. destruens* in that they have mature spherical sporangia (the enclosure in which spores are produced asexually) between 40 and 400 µm in diameter that release infective spores through a pore. The released spores then increase in size until they become mature sporangia containing hundreds of spores and the cycle begins again. Members of the genus *Rhinosporidium* cause disease in humans, dogs, cattle, horses, and swans that is characterized by the formation of polyps, usually on the mucous membranes of the nose or nostrils, eyes, and mouth, with a chronic granulomatous inflammatory response consisting of mononuclear cells, polymorphic nuclear cells, and, in some cases, giant cells [51]. Therefore, infection by *S. destruens* differs from that of members of *Dermocystidium* and *Rhinosporidium* since the spore stages are found as intracellular parasites of host cells, infecting and replicating mainly in visceral organs where they cause a chronic granulomatous disease and only produce zoospores when released into freshwater [26].

When compared with more phylogenetically distant fungal pathogens, such as the well known *Batrachochytrium dendrobatidis* that causes chytridiomycosis in amphibians, the life cycle and mechanisms of the pathogenicity of *S. destruens* appear to be very different, although they exhibit intracellular parasitism. *Batrachochytrium dendrobatidis* compromises epidermal tissues of bullfrogs by colonizing keratin-containing cells [2]. Pathological signs of chytridiomycosis include accumulation and erosion of corneal cells (epithelial and endothelial cells, keratocytes), swelling of the epidermis, damaged nuclei, and altered cytoplasm [2]. Death of the animal is caused by inhibition of electrolyte transport across the epidermis, followed by disruption of cardiac electrical activity [52]. The asexual life cycle of this fungus consists of a motile zoospore stage and a stationary thallus stage [53]. The zoospores are flagellated and are not bound by a cell wall. Once exposed to a frog host, a zoospore encysts (attaches itself, retracts its flagellum, and forms a chitin wall around the spore body) on the host skin surface and produces a germination tube that penetrates the host epidermis. The content of the cyst then migrates into the host tissue via this tube and a single or colonial zoosporangium. A septum forms to separate the zoosporangium from the tube and maturation of the zoosporangium results in the cleavage of the zoosporangial contents into zoospores, which are then released from the host cell. Infected cells show, among other things, displacement of host organelles and clear zones around the zoosporangia [2]. Although these examples are not exhaustive, they suggest that the life cycle of *S. destruens* and the mechanisms of pathogenicity in infected fish hosts are specific to this parasite and also highlight the need for further study of its mechanisms of entry into cells, its modes of replication in infected cells, and the direct infectivity of zoospores.

### 1.6. All Species and Ontogenetic Stages Are Not Equally Susceptible to S. destruens

When looking at the prevalence of *S. destruens* in populations of different susceptible host species, there is a large variability between species. Prevalence is measured here by the number of PCR-positive individuals with the rosette agent in all the fish tested within a population. We therefore observe prevalences that fluctuate from 100% to 2% depending on the species tested in the cyprinidae family, and from 98% to 3% in salmonids (Figure 4).

Using the RA-3 isolate, Arkush et al. [26] experimentally tested the susceptibility of a variety of juvenile salmonids. While their experimental infections led to a clinically identical disease to that previously described by Harrell et al. [23] in *O. tshawytsch* from Washington (USA), the authors found a trend towards host specificity, with *O. tshawytsch* being the most susceptible species to *S. destruens*, followed by coho salmon *Oncorhynchus kisutch*, rainbow trout *Oncorhynchus mykiss*, brown trout *Salmo trutta*, and brook trout *Salvelinus fontinalis*, the latter showing resistance to infection by *S. destruens* [26]. In addition, Gozlan et al. [27,28] conducted cohabitation experiments in both natural ponds and laboratory conditions using *P. parva* captured from wild pond populations in Hampshire, UK, originally introduced in 1985 from German Danube populations. Their results showed up to 69% mortality of *L. delineatus* under laboratory conditions and up to 96% mortality of fish in natural ponds. By implementing the first PCR-based DNA detection tool targeting a small fragment of ribosomal DNA (18S rRNA gene) [27] and then the Internal Transcribed Spacer 1 (ITS-1) [28] of *S. destruens*, they were able to estimate the prevalence of the disease to be 67% in *L. delineatus* and 20% in the fathead minnow *P. promelas*. Conventional bacteriological, virological, parasitological, and histological examinations were carried out on moribund *L. delineatus* fish and showed extensive infections of visceral organs by an intracellular parasite with characteristics similar to those of the previously described rosette agent in *O. tshawytscha* [23,26] and *S. salar* [25]. Spikmans et al. [30,31] found that *P. parva* populations sampled in the Meuse floodplain were infected with a disease prevalence of up to 67% at the Everlose Beek site and 74% and 25% for the Teelebeek site. In addition to the invasive *P. parva* populations, a disease prevalence of 25% was found in the native *Gasterosteus aculeatus* (*Gasterosteidae*). In Turkey, in the Sariçay stream in Mugla, *S. destruens* was found by PCR in *Oxynoemacheilus sp.* (*Nemacheilidae*, prevalence 67%), *P. smyrnaeus* (prevalence 100%), *S. fellowesii* (prevalence 67%), *D. labrax* (*Moronidae*, prevalence 33–100%), and *L. gibbosus* (*Percidae*, prevalence 75%). Based on these results, Ercan et al. [32] highlighted the potential role of the emergence of *S. destruens* in the extinction events observed in Turkey since 2009 in *Oxynoemacheilus sp.*, *P. smyrnaeus*, and *S. fellowesii*. Finally, in France, *S. destruens* was detected in populations of *P. parva* (prevalence 2–4%), bleak *Alburnus alburnus* (prevalence 9%), European bittern *Rhodeus amarus* (prevalence 20%), roach *Rutilus rutilus* (prevalence 4%), gudgeon *Gobio gobio* (prevalence 10%), and minnow *Phoxinus phoxinus* (prevalence 2%) [22]. The *P. parva* populations had a low parasite prevalence and did not develop the disease.

It is therefore very important in an infectious risk analysis following the arrival of a healthy *P. parva* carrier to take into account the species present in the community. However, one hundred percent of the species tested or analyzed for mortalities related to *S. destruens* were found to be susceptible to this infectious agent. This also applies to percids, such as pumkindseed *L. gibbosus*, or even marine species, such as sea bass *D. labrax*, as studies have shown that *S. destruens* spores survive very well in both salt and freshwater [11]. In view of the host spectrum within this new group of emerging infectious agents (i.e., *Dermocystida*), it would be useful to test the susceptibility of amphibians and invertebrates, such as certain crayfish, to the rosette agent to see if we are indeed dealing with an infectious agent of fish. Indeed, the pathogenic fungi of this group affect fish as well as amphibians and humans [16,17,19] and therefore, the generalist aspect of *S. destruens* should be tested beyond the simple fish group.

It is also important to note that host species with short life spans and rapid reproductive systems such as *L. delineatus*, *R. amarus*, or even *L. gibbosus* that depend on high reproduction to maintain populations will show a more visible decline more quickly. In effect, the emergence and high mortalities of *S. destruens* in *O. tshawytscha* stocks have occurred in juvenile or smolting salmons and not in brood stocks [26]. In contrast, *S. destruens* was found in 30% of returning salmon, showing that they survived infection (asymptomatic infections, [11]), which would suggest that species or stages with larger body mass hosts would be more resistant to infection in the short term.

### 1.7. What Is the Potential Economic Impact of the Emergence of S. destruens?

As mentioned, the first emergences of *S. destruens* were observed in salmon farms in the USA with losses of up to 98% of juvenile salmon stocks. In such a situation, assessment of the economic cost is fairly straightforward as each fish has an economic value defined by aquaculture production in a specific national context. Where the economic impact is more complicated to assess is in the context of wild populations. In their recent assessments of the economic costs of invasive species, Diagne et al. [54] show that the costs of introducing new pathogens are often assessed indirectly, either through the management costs of susceptible species or through the costs of eradicating the host. However, there is still very little data on the estimated costs of infectious agents per se and none on *S. destruens*. However, environmental agencies in general assess that the primary risk of impact from the introduction of *P. parva* on native fish populations is essentially the infectious risk from the emergence of *S. destruens*. Based on the data on the emergence of *S. destruens* in the absence of the healthy carrier *P. parva*, it is reasonable to assume that the risk posed by *S. destruens* would no longer be endemic, as has occurred in the USA. In this context, England has been a pioneer in managing this risk with the establishment of a program focused on rapid detection and eradication of populations of *P. parva* carrying *S. destruens* [55]. Britton et al. [56] accurately estimated the cost of a rotenone-based eradication and containment strategy for the management of *P. parva*. The cost of eradicating a population of *P. parva* averaged €80k over ten years, with 90% of the costs being for the first year of eradication and the remaining 10% being residual costs related to post-eradication monitoring. On this basis, which remains approximate but nonetheless quantitative and specific to the eradication program for populations of the healthy carrier *P. parva*, it is therefore possible to evaluate the economic cost of controlling *S. destruens* throughout the European continent based on the number of *P. parva* populations present in each country. This gives an average cost per country of just over 85M€, with a minimum cost of about 92k€ for Bosnia and Herzegovina and a maximum cost of about 43M€ for Germany. The total cost of eradicating the healthy carrier of *S. destruens* in Europe would be around 188M€ (Figure 5). Thus, it is quickly realized that, from an economic point of view, the introduction of *S. destruens* into an ecosystem becomes very quickly prohibitive. The total cost of eradicating the healthy carrier in England was around €15 million as angling in that country represents a huge economic market; larger than the football Premier League, to put it mildly. Each country has its own situation and a cost-benefit analysis must be carried out before any eradication program can be implemented. Above all, we must learn from this case study that eradication is the last option and that other solutions for managing an *S. destruens* epidemic are possible.

### 1.8. What to Do When the Emergence of S. destruens Is Identified in a Water System?

A first conclusion is that the rosette agent *S. destruens* is indeed present in several countries in Asia and Europe in several locations. It has been found to be associated with populations of *P. parva* and native cyprinid and salmonid species have been identified as carriers [57]. From what is known about this infectious agent and its mode of transmission, it is likely that high local and national fish kills could be attributed to the emergence of this disease. With the hindsight we now have, there is no reason to believe that *S. destruens* will reach a level of equilibrium with local species that will allow for long-term cohabitation with *P. parva* healthy carrier populations [58]. Indeed, the presence of a healthy carrier prevents any balance between the virulence of the infectious agent and the vulnerability of the native species present (in contrast to what happened in the salmon farms in California). Another conclusion is that *P. parva* and *S. destruens* are not yet everywhere and therefore, there are areas and rivers (e.g., headwaters) that deserve greater protection against future introductions. Finally, a very important point to remember is that we do not yet have quantitative data on the diversity of strains present and their virulence. Indeed, it is likely that all the identified strains of *S. destruens* are different on a genetic level, as previously shown on a global scale by Sana et al. [37]. It would therefore be urgent to test the virulence of the strains identified in Europe and Asia in order to isolate exchanges with populations carrying these strains and thus be able to prioritize management of the infectious risk.

### 1.9. Recommendations

At this stage of our knowledge, it is important to take the infectious risk linked to *S. destruens* very seriously. All studies carried out to date point to rapid declines in fish populations following the introduction of *S. destruens* into fish farming communities. The first recommendation is the isolation of *P. parva* populations by controlling the transfer of fish from infected areas to non-infected areas. *S. destruens* can be transferred by free spores in freshwater, by transport of infected native fish, and also by transfer of infected *P. parva*. Of these three modes of contamination, the last is the most serious as it allows, in addition to introduction of the pathogen, the introduction of the healthy carrier which will serve as a reservoir for *S. destruens* and thus maintain a high level of virulence. These controls must be carried out for any transfer of fish to an area where *P. parva* is not yet present. There have been studies in England undertaken by the Environment Agency which tested the effectiveness of stocking controls and associated procedures against stocking levels of *P. parva* contamination ranging from 1, 5, 10, and 20% and the level of expertise of the auditors ranging from expert, intermediate, and novice. They showed that, with contamination rates of 10%, the probability of detection of *P. parva* by a trained team was above an 80% probability of detection. Such a study could be adapted to the French stocking management system.

The second recommendation is public risk communication. Indeed, the challenge of this approach is to include the maximum number of actors in the sector in this fight. This includes professional fish farmers, pond owners, and amateur fishermen. The greater the number of people informed of the risks linked to the introduction of *P. parva* and *S. destruens*, the more the authorities responsible for environmental protection will have allies in the field who will be involved in this fight. This will be done through communications within fishing federations, letters to fishing license holders, posters in fishing shops, and communications to fish farmers and other professionals in the sector. The use of social networks as a new means of communication is also highly recommended. The more information is shared, the more responsibility there will be on the part of the various actors. It is obvious that this will not be enough, but as many studies have shown, the alternative of not effectively communicating slows down the fight against biological invasions, such as *P. parva*. This recommendation must be part of a national coordination and implementation of a common strategy for the whole territory.

The third recommendation is to analyze the virulence of the strains found. Indeed, it is very likely that the strains found in a country are the result of their introduction history and are therefore not the same from a genetic point of view. More importantly, the virulence of these strains in relation to populations of native species is likely to be highly variable. The cultivation of these different strains would therefore be an important first step in testing their virulence. Their inoculation on different cell lines (e.g., *Salmonidae*, *Cyprinidae*) would be a simple and inexpensive indicator of virulence.

The fourth recommendation is the targeted eradication of *P. parva* populations. This is the option that was chosen by the UK government in 2005, which, through the use of rotenone on sites hosting *P. parva* populations, reduced the number of *P. parva* populations in the country from 37 to six in 2014 and targeted total eradication by 2017. According to the latest reports, they should have achieved their target. It is an expensive program but given the economic importance of freshwater recreational fishing in the UK, it has received the necessary budgetary support. In any country, a limited and targeted approach to the most economically or conservationally sensitive sites could therefore be an option for concerned organizations. A derogation request for the use of rotenone will nevertheless be mandatory at the European level.

The fifth recommendation is the search for new tools. Indeed, new scientific tools could be developed for both the detection and eradication of *P. parva* populations. First of all, the use of environmental DNA to monitor the presence of invasive species in a body of water or in a catchment area is already being studied in other countries. By sampling water, a list of present species can be established and thus allow for the routine and low-cost monitoring of communities and, above all, the rapid identification of the arrival of an invasive species. This would allow for early detection and eradication at the very beginning of invasion with a much greater chance of success and also, a reduction in the costs of eradication. Other tools such as the introduction of *P. parva* genetically modified to produce only males could be considered, at least in lakes or ponds. This kind of approach has already been initiated in Australia on carp populations and in Africa on mosquito populations. It would also be possible to work on similar approaches for *S. destruens* by introducing a deleterious gene with more limited ethical issues and environmental risks.

In summary, firstly, there is a priority in the implementation of recommendations and in the isolation of *P. parva* populations to minimize the risk of transfer from one body of water to another. Secondly, discussions will have to take place to evaluate the infectious risk linked to *S. destruens* in a wider context of other infectious risks on wild and farmed fish populations (i.e., inclusion on the OIE’s aquatic animal health code). We would therefore like to note that *S. destruens* is not specific to one species, but it always leads to mortality in a whole range of species and families of fish including *Cyprinidae*, *Salmonidae*, *Percidae*, and marine species, such as sea bass. We would also emphasize that it is the presence of the healthy carrier that maintains the high level of virulence.

## Figures and Tables

**Figure 1 jof-09-00426-f001:**
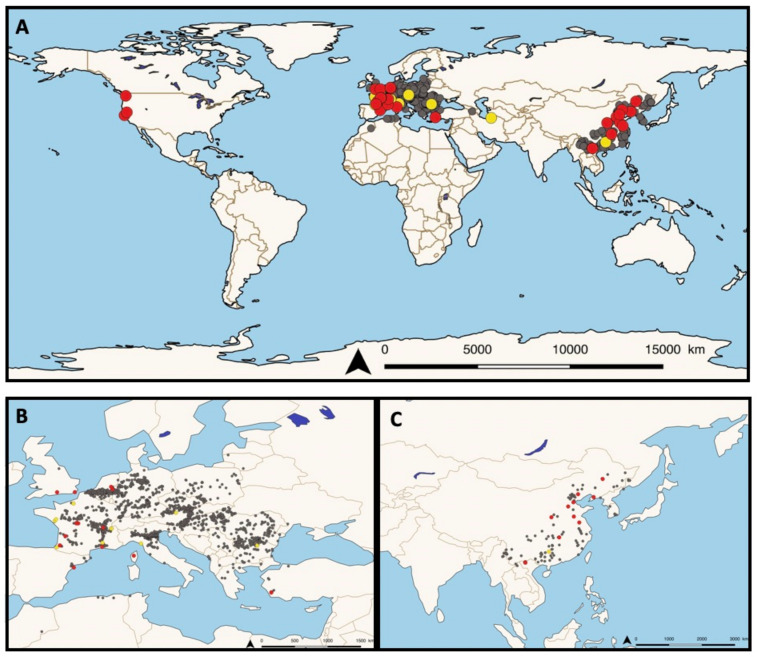
(**A**) Worldwide distribution of the generalist fungal parasite *S. destruens*. The distribution of the asymptomatic carrier host *Pseudorasbora parva* in its native (Est Asia) and invasive range is indicated by grey dots. The red and yellow dots indicate the distribution of *P. parva* and other native species tested for the presence of the agent rosette *Sphareothecum destruens* DNA; Red dots indicate populations found to be positive to *S. destruens* DNA and yellow dots indicate non-conclusive results. (**B**) Invasive distribution and (**C**) native (China) and invasive Vietnam distribution in Asia.

**Figure 2 jof-09-00426-f002:**
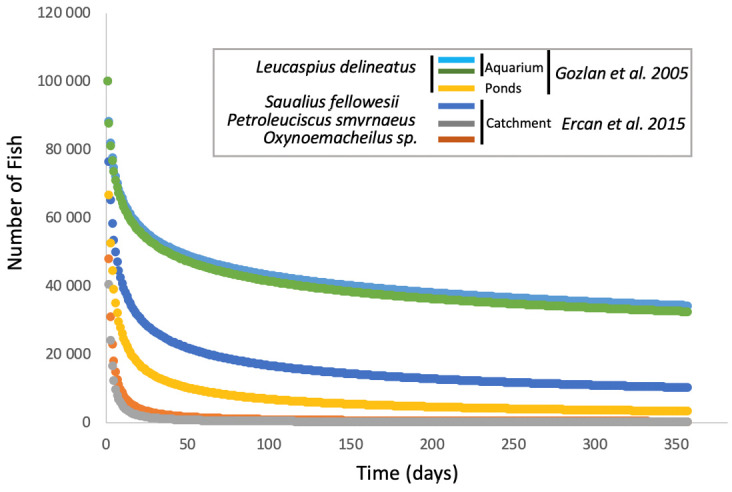
Mortality curves after infection with the rosette agent *Sphareothecum destruens* in different species and under different environmental conditions (based [27,32] for data and [39] for model).

**Figure 3 jof-09-00426-f003:**
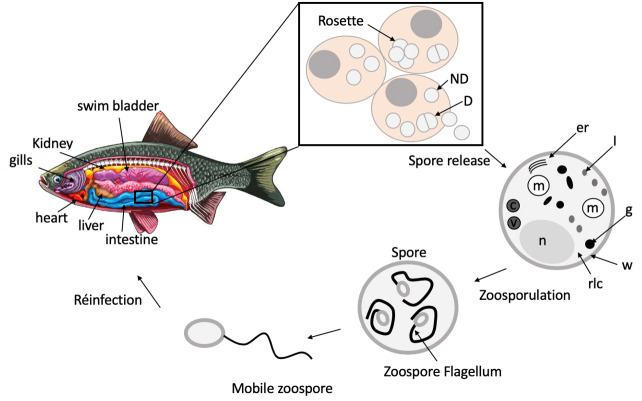
Life cycle, spore stages, and ultrastructure characteristics of RA. In infected fish, RA spores are observed in various organ tissues and cell hosts, including spleen, kidney, liver, intestine, heart, gills, and swimbladder, among others. Some of these are represented here. In fish tissues, RA spores are mainly intracellular and intracytoplasmic, although some spores are found extracellularly. Within infected host cells, single spores are present and various aggregates of apposed spores are also observed as rosette structures. Different spore stages are found, such as dividing (4–6 µm in diameter) and non-dividing stages (2–4 µm in diameter). Each spore can produce up to five daughter spores. Death of host cells allows for RA spores to be released into freshwater where they undergo zoosporulation via the uncoiling of a motile flagellum that comprises a body of approx. 2 µm in diameter and a flagellum of approx. 10 µm in length. RA spore ultrastructure; D: dividing spore stage; ND: non-dividing spore stage; er: endoplasmic reticulum; I: lipid droplets; g: electron dense granules; w: trilaminar cell wall with fibrogranular material; rlc: ribosome-laden cytoplasm; n: nucleus; v: vacuole; c: vacuole with concentric bodies; m: mitochondria.

**Figure 4 jof-09-00426-f004:**
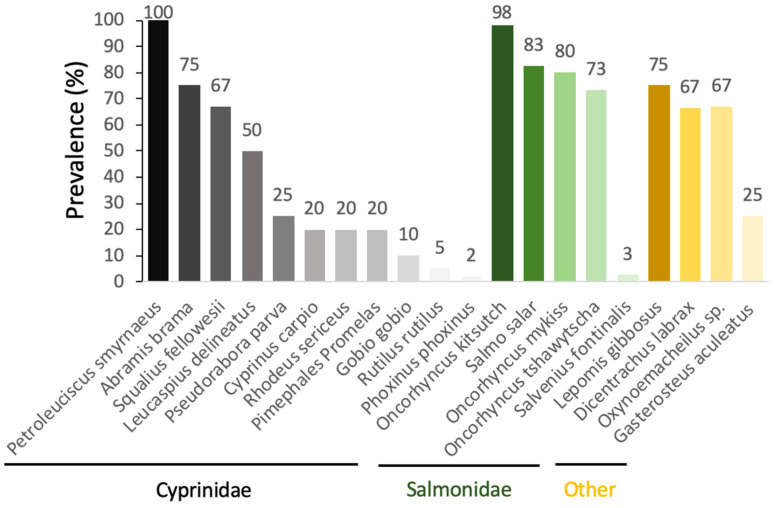
Prevalence of the rosette agent *Sphareothecum destruens* detected by PCR in different species of wild and aquacultured fish.

**Figure 5 jof-09-00426-f005:**
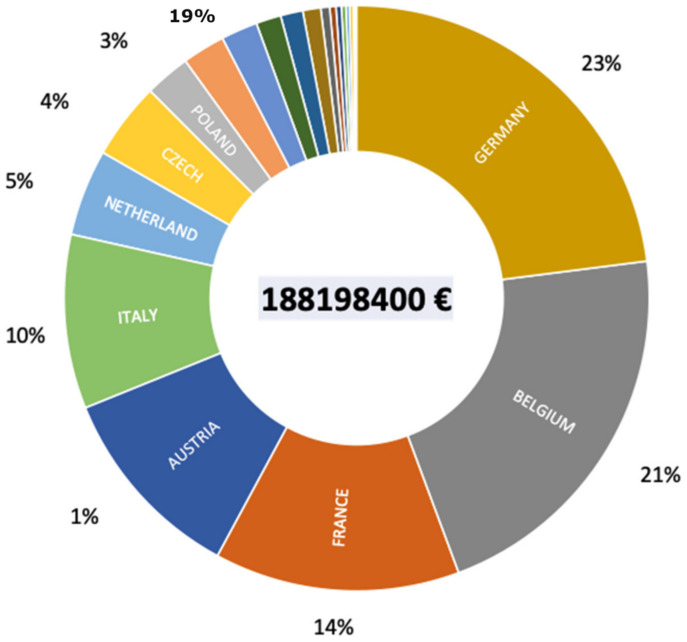
Total estimated cost in € for eradicating populations carrying the rosette agent *Sphareothecum destruens* in different European countries with a breakdown per country of this cost according to the number of populations present in each country (Britton et al., 2011, [56]).

**Table 1 jof-09-00426-t001:** Description of organ and tissue types infected with RA, the histopathological features of the disease, as well as RA spore dissemination within tissues.

Fish Species	Infected Organs/Tissues	Histopathology of Infected Tissues	RA Dissemination within Tissues	References
Chinook salmon (*O. tshawytscha*)	Spleen, kidneys	Focal areas of RA spore growth, necrosis	Intracellular localization of spores in macrophages and endothelial cells	[24]
Chinook salmon (*O. tshawytscha*)	Spleen, kidneys	Edema, focal necrosis	Spherical organisms of 2–7 µm in diameter with a peripheral halo that occurred in cluster “rosette”, organisms accumulate in macrophages, intracellular organisms found within the interstitium parenchyma	[23]
	Spleen, kidney, liver, gonad, heart, brain, intestinal mucosae	Hepatomegaly, splenomegaly	Organisms observed in peripheral blood and vascular spaces of these organs	
Atlantic salmon (*Salmo salar*)	Spleen, kidney, liver, gonads	Widely disseminated nodules, with involvement of hematopoietic tissues	Spores of 2–7 µm in diameter found principally in macrophages but also as cell-free forms	[25]
	Spleen	Granulomas in splenic and hepatic lesions with macrophages at the periphery of the lesions		
	Spleen, kidney, testes	More diffuse granulomatous response		
Winter-run chinook salmon (*O. tshawytscha*)	Spleen, kidney, liver, heart, mesentery surrounding the intestinal tract, pyloric cecae	Nodular form: multifocal granulomas that replaced the normal parenchyma, nodules observed in visceral organs. Granulomas characterized by central cores of eosinophilic necrotic material or closely apposed macrophages	Aggregates of RA found within central zones of granulomas and within macrophages	[11]
	Spleen, kidney, liver, heart, gill, brain, ovary, testis, hindgut	Disseminated form: edema, focal necrosis, enlargement and pallor of the spleen kidney, liver	RA spores found in hematopoietic, epithelial and mesenchymal cells, as intracellular or extracellular forms, clusters of 4–5 rosettes	
	Kidney	Necrosis of the renal tubular epithelium, loss of tubules, membranous glomerulonephritis, necrotizing interstitial nephritis	Parasite present as single or in aggregates within the cytoplasm of the bilary, renal tubular epithelium, lumina of bile ductules and renal tubules	
	Spleen	Necrotizing vasculitis of splenic arterioles	RA spores largely disseminated individually or in aggregates in the pulp spaces, in the cytoplasm of sinusoidal macrophages and reticuloendithelial cells. RA spores found in the lumina and tunicae media of splenic arterioles	
	Gills (in early infections)		RA spores found within vessels of the gill	
	Swimbladder (in advanced infections)		RA spores found in subserosal aggregates	
	Epidermis, urine, seminal and ovarian fluids, intestine mucosa		RA sometimes observed	
Sunbleak (*L. delineatus*)	Spleen, kidney, liver, intestine, gonad, eye, adipose tissue surrounding the intestinal tract, skeletal tissue	Nodular form Disseminated form Vacuolar degeneration, necrosis	RA spores located intracellularly in various types of host cell, including renal tubule and collecting-duct epithelial cells. Presence of spores within giant cells.	[45]
	Kidney, testis	Intense inflammation	Numerous stages of RA spore, mostly intracellular in the nodular form, intracellular and extracellular form for disseminated disease, aggregates “rosette”	
	Liver	Inflammatory response resulting in an influx of phagocytic cells, lymphocytic infiltration of the hepatic parenchyma. Multifocal granuloma of different size		
	Eyes		RA spores within macrophages	
	Testis	Multifocal granuloma of different size, necrosis and intense inflammation		
Sunbleak (*L. delineatus*)	Kidney, liver, testis, gill	Nodular form: multifocal granulomas in liver and testis Disseminated form: Hepatocellular necrosis of the liver	Granulomas enclosed different stages of RA spores, 2–4 µm in diameter Intracellular and extracellular RA spores	[28]
Atlantic salmon (*Salmo salar*)	Spleen, kidney, liver, heart, choroidal rete, cranial connective tissue	Inflammatory lesions		[46]
	Kidney	Granulomatous lesions surrounded by hepatocytes and inflammatory cells	Proliferation of RA spores in haematopoietic tissues, RA spores within macrophages	[46]
	Liver	Numerous plaques of pale tissues and extensive inflammation	RA spores of different sizes	[46]
	Spleen	Numerous plaques of pale tissues and extensive inflammation		[46]
Sunbleak (*L. delineatus*)		Pancreatitis, severe inflammation of the spleen and renal interstitial haematopoietic tissue	Intense proliferation of RA spores	[46]

Brief definitions: Macrophage: a large white blood cell in the immune system involved in the detection, phagocytosis, and destruction of harmful organisms. Granuloma: an aggregation of macrophages that forms in response to chronic inflammation, which occurs when the immune system attempts to isolate foreign substances that it is otherwise unable to eliminate. Necrosis: a form of cell injury which results in the premature death of cells in living tissue by autolysis. Hematopoietic tissue: tissue in which new blood cells are formed, notably the bone marrow, the lymph nodes, and the spleen, which allow for the formation of blood cells via hematopoiesis. Parenchyma: the functional tissue of an organ as distinguished from the connective and supporting tissue. Nodule: tumor formed by a cluster of cells (hepatocytes in the case of hepatic nodule). Edema: swelling of an organ or tissue due to an accumulation of fluid in the interstitial medium. Hepatomegaly: enlarged liver. Splenomegaly: enlarged spleen. Eosinophilic: the staining of tissues, cells, or organelles after they have been washed with eosin, a dye.

## Data Availability

Not applicable.

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
