# Peer review of "Emergence of the Fungal Rosette Agent in the World: Current Risk to Fish Biodiversity and Aquaculture"

_jof, 2023, doi:10.3390/jof9040426_

Round 1

Reviewer 1 Report

Journal of Fungi

Manuscript Number: JoF-2201528

Title: Emergence of the fungal rosette agent in the world: Current 2 risk to fish biodiversity and aquaculture.

Reviewer

General Comment:

The manuscript is a review dealing with an account of the current knowledge about the dermocystid fungus Sphaerothecum destruens, including its distribution, detection and prevalence. The review also presents data on the associated mortality in wild and farmed fish species, and the potential economic impact in countries where the healthy carrier, the topmouth gudgeon Pseudorasbora parva has been introduced. The review also presents some perspectives to manage and mitigate the presence of the pathogen in the regions where it is established.

The manuscript is interesting and provides an up-to-date account of the status of the disease and prersence of the pathogen in fish farming countries, especially those of salmonids. It is advised to make a thorough check of English throughout the manuscript. Specific comments are given below:

Specific comments

Introduction

Comment 1 line 30. The word “shown” is duplicated.

Comment 2 lines 52-54. The genera and species names should be in italics.

Comment 3 line 71 should say: “…other enigmatic parasites of fish and crustaceans”

Comment 4 line 72 should say: “…were the first to obtain the complete DNA sequence…”

Comment 5 line 79. The author of reference [16] should be mentioned for clarity.

Comment 6 title section 1. It is suggested to change the title of the section to make it shorter and concise.

Comment 7 line 147. Delete the word “was”

Comment 8 line 201. Add the author name before the citation [37] to improve clarity.

Comment 9 section 1.2 title is suggested to be changed as follows:

1.2. Mortalities associated with the emergence of S. destruens

Comment 10 line 224 should say: “The parasite Sphareothecum destruens causes low-level mortalities which…”

Comment 11 line 238. It seems that citations are missing in the sentence. Please add such citations.

Comment 12 lines 243 and 263. Make sure about the meaning of the word “catchment” in the sentences. It is suggested to change the word “chatchment” at least in line 263 for the word “fishery”.

Comment 13 line 272. It also should be mentioned the role of the healthy carrier on the pathogen persistence in the aquatic system.

Comment 14: It is suggested to describe the events following the introduction of the healthy carrier to a water body where other susceptible fish species exist, related to the infection and spread of the pathogen S. destruens

Comment 15: It is suggested to include a table with the existing PCR methods and primers used to detect the pathogen.

Comment 16 line 422. Duplicate citation [11].

Comment 17 line 423 and 430. Dermocystidium should be in italics.

Comment 18 section 1.6 line 472. The title of the section should be concise regarding host and development stage susceptibility differences to S. destruens.

Comment 19 line 484 and elsewhere. Make sure all genus and species names are in italics.

Comment 20 figure 4. It is adviced to use a different color for each fish family and then use different tones or shades of the color for each species of the family, in order to make it uniform and colorful.

Author Response

Specific comments

Introduction

Comment 1 line 30. The word “shown” is duplicated.

Duplication has been removed

Comment 2 lines 52-54. The genera and species names should be in italics.

Done

Comment 3 line 71 should say: “…other enigmatic parasites of fish and crustaceans”

Done

Comment 4 line 72 should say: “…were the first to obtain the complete DNA sequence…”

Done

Comment 5 line 79. The author of reference [16] should be mentioned for clarity.

 Done

Comment 6 title section 1. It is suggested to change the title of the section to make it shorter and concise.

 Done

Comment 7 line 147. Delete the word “was”

 Done

Comment 8 line 201. Add the author name before the citation [37] to improve clarity.

 Done

Comment 9 section 1.2 title is suggested to be changed as follows:

1.2. Mortalities associated with the emergence of S. destruens

Done

Comment 10 line 224 should say: “The parasite Sphareothecum destruens causes low-level mortalities which…”

Changed

Comment 11 line 238. It seems that citations are missing in the sentence. Please add such citations.

Citations added line 237

Comment 12 lines 243 and 263. Make sure about the meaning of the word “catchment” in the sentences. It is suggested to change the word “chatchment” at least in line 263 for the word “fishery”.

Word changed line 275

Comment 13 line 272. It also should be mentioned the role of the healthy carrier on the pathogen persistence in the aquatic system.

This point was developed in the manuscript

Comment 14: It is suggested to describe the events following the introduction of the healthy carrier to a water body where other susceptible fish species exist, related to the infection and spread of the pathogen S. destruens

Done. We have now detailed lines 252-260 of the events following the introduction of the healthy carrier into a body of water with susceptible fish.

Comment 15: It is suggested to include a table with the existing PCR methods and primers used to detect the pathogen.

We recently published a paper that focuses on the PCR methods to detect S. destruens DNA where a table and a figure indicating the different primer pairs are available. It seems thus redundant to include this in the present review.

See Cherif et al. (2022) Assessing the specificity of the Rosette agent DNA amplification: an optimized protocol for the detection of standard DNA among studies. Journal of Fish Diseases. DOI: 10.1111/jfd.13722

Comment 16 line 422. Duplicate citation [11].

Removed line 436

Comment 17 line 423 and 430. Dermocystidium should be in italics.

Done

Comment 18 section 1.6 line 472. The title of the section should be concise regarding host and development stage susceptibility differences to S. destruens.

Corrected

Comment 19 line 484 and elsewhere. Make sure all genus and species names are in italics.

Verified

Comment 20 figure 4. It is adviced to use a different color for each fish family and then use different tones or shades of the color for each species of the family, in order to make it uniform and colorful.

Done. We have colour coded each family of fish and used different colour tones for each species.

Reviewer 2 Report

Very interesting and useful. Would you please consider introduction proposal (concept) of future study on anti-infectice strategies related to reson of          the presence of the healthy cariers ?

Author Response

Very interesting and useful. Would you please consider introduction proposal (concept) of future study on anti-infectice strategies related to reson of          the presence of the healthy cariers ?

Five recommendations or strategies have been suggested to limit the introduction and subsequent impact of P. parva as a healthy carrier. We have also proposed avenues for future research to control the introduction and emergence of this disease. Line 677-689.

Reviewer 3 Report

The subject is interesting and worth studying. I think it is worth the publication whereas, manuscript needs to correct some small details to further enhance the quality of the manuscript. English needs to be improved significantly. 

Author Response

The subject is interesting and worth studying. I think it is worth the publication whereas, manuscript needs to correct some small details to further enhance the quality of the manuscript. English needs to be improved significantly. 

Some changes were done according to the reviewer’s comments. English has been checked by a native speaker.